# Values and Diagnostic Accuracy of Electrodiagnostic Findings in Carpal Tunnel Syndrome Based on Age, Gender, and Diabetes

**DOI:** 10.3390/diagnostics14131381

**Published:** 2024-06-28

**Authors:** Ahmad R. Abuzinadah

**Affiliations:** 1Department of Neurology, Faculty of Medicine, King Abdulaziz University, Jeddah 21589, Saudi Arabia; aabuzinadah@kau.edu.sa; 2Neuromuscular Medicine Unit, King Abdulaziz University Hospital, King Abdulaziz University, Jeddah 21589, Saudi Arabia; 3Department of Internal Medicine, Neurology Division, International Medical Center, Jeddah 23214, Saudi Arabia

**Keywords:** carpal tunnel, diagnosis, diabetes, screening, predictive values

## Abstract

Background: Appropriate cut-off values and diagnostic accuracy (DA) [sensitivity, specificity, predictive values, positive (PPV) and negative (NPV)] of electrodiagnostic findings for carpal tunnel syndrome (CTS) based on age, gender, and diabetes mellitus (DM) were not reported. Methods: In a retrospective study, we described the latency difference values and DA of comparative latency studies COLS [median to ulnar through palmar difference (palmdiff) and ring difference study (ringdiff), median to radial through thumb difference study (thumbdiff), and combined sensory index (CSI)] among non-CTS and CTS groups based on age, DM, and gender. Results: We included 632 patients (228 without CTS and 404 with CTS). For PPV > 90% and NPV > 50%, the best cut-offs among patients without DM, were 0.5ms, 0.8–1 ms, 1.4 ms, and 2 ms for palmdiff, thumbdiff, CSI (age < 60 years), and CSI (age > 60 years), respectively. The best cut-offs among patients with DM were 0.5 ms, 1.2 ms, 0.8 ms, 1.0–1.2 ms, 1.8 ms, 1–1.2 ms, 3.0 ms, and 3.5 ms for palmdiff (age < 50 years), palmdiff (age > 50 years), thumbdiff (age < 40 years), thumbdiff (age 40–59 years), thumbdiff (age > 60 years), CSI (age < 50 years), CSI (age 50–59 years), and CSI (age > 60 years), respectively. Conclusion: The cut-off values of COLS to confirm CTS and their DA were different according to age and DM.

## 1. Introduction

Carpal tunnel syndrome (CTS) is a prevalent neuropathic disorder marked by the entrapment of the median nerve within the carpal tunnel, manifesting clinically with symptoms such as tingling, numbness, pain, and, in severe cases, weakened grip strength. Initially identified studies reported a general population prevalence of 3.8%, with lifetime occurrence rates reaching up to 10% [1,2]. Nerve conduction studies (NCSs) and, in particular, the latency of sensory potentials, are used to confirm the diagnosis of CTS because of the high specificity of NCSs for CTS diagnosis [3,4]. While NCSs are the gold standard for CTS diagnosis due to their high specificity, emerging modalities, such as ultrasound, are gaining favor for their supportive diagnostic capabilities [2,5]. However, NCSs remain superior in differentiating between axonal and demyelinating median neuropathies, which is important for deciding between conservative management and surgical intervention [6]. This differentiation is particularly vital in severe CTS cases, especially among patients with diabetes mellitus (DM), where delayed surgical intervention after axonal loss occurs might not yield significant benefits [7]. Subsequently, it is important to make a CTS diagnosis before axonal loss occurs. CTS also coexists with many other musculoskeletal causes of hand pain, such as rheumatoid arthritis and fibromyalgia, and the presence of confirmatory tests is important in providing timely care [8].

Despite extensive research regarding the sensitivity and specificity of NCSs in diagnosing CTS, three gaps remain in the literature that need to be addressed [3]. First, the available literature does not report positive and negative predictive values (PPV, NPV) among the non-selective population. PPV predicts the chance that a patient has the disease if a test result is positive, while NPV predicts the chance that a patient does not have the disease if a test result is negative. PPV and NPV are highly dependent on the actual prevalence of the disease [9]. The accurate calculation of PPV and NPV requires data from a nonselective cohort to avoid skewing disease prevalence. Robinson et al. reported PPV and NPV among 54 patients with CTS versus 46 control. Since their control cases were selected, this may not have reflected the broader population prevalence [10]. Second, prior studies did not report sensitivity and specificity based on age. It was estimated that the median nerve sensory peak latency increased by 0.11 ms over a five-year period, which highlights the need to consider patient age when interpreting CTS electrodiagnostic studies [11,12]. Third, prior studies did not report sensitivity and specificity based on the presence of DM, except for studies with a small sample size (e.g., 28 patients) [13].

The aim of our study was to report the best cut-off values and the diagnostic accuracy (DA) of comparative latency studies (COLSs) to diagnose carpal tunnel syndrome in a nonselective cohort that represented the prevalence of CTS among the clinical neurophysiology lab (CNPL) population in a whole cohort and based on age, gender, and the presence of DM.

## 2. Materials and Methods

### 2.1. Study Design and Participants

Data were collected from a chart review of prospectively collected data between January 2017 and July 2023 at King Abdulaziz University Hospital (KAUH) and the International Medical Center (IMC). The KAUH and IMC institutional review boards approved the protocol. Our purpose was to include a population that represented a real clinical practice of patients who were referred to the clinical neurophysiology lab (CNPL) with upper symptoms to rule in or rule out carpal tunnel syndrome. Hence, our cohort included patients with musculoskeletal diseases (MSKDs), ulnar neuropathy, and cervical radiculopathy to serve as a control group, as these commonly include CTS as part of their differential diagnosis. Our inclusion criteria were as follows: (1) age range between 14 and 80 years; (2) presenting to CNPL with upper limb paresthesia or pain; and (3) carpal tunnel syndrome among the possible differential diagnoses. The exclusion criteria included (1) weakness outside the hands or carpal tunnel syndrome distribution and (2) cases in which we were unable to classify them into either group (CTS group or non-CTS group) due to the coexistence of possible carpal tunnel syndrome clinical features and MSKD symptoms and physical signs. MSKDs included carpometacarpal arthritis (CMC), de Quervain’s tenosynovitis (DTS), trigger fingers, lateral epicondylitis (tennis elbow), medial epicondylitis, rotator cuff syndrome, myofascial pain, and fibromyalgia.

### 2.2. Variable Definitions

Carpal tunnel syndrome cases were defined by fulfilling two of the following three features: (1) nocturnal paresthesia; (2) the aggravation of paranesthesia by activities such as driving a car, riding a bike, holding a book, or holding a telephone; and (3) paresthesia relieved by shaking the hand (positive flick sign) [14,15]. Several investigators have suggested that clinical diagnosis remains the gold standard for diagnosing carpal tunnel syndrome [16,17]. We used the above criteria as it has shown to predict benefits from CTS release in randomized trials. These criteria did not include median nerve distribution, as it has been reported that more than one-third of CTS cases do not follow median nerve distribution [18]. Median neuropathy electrodiagnostic parameters were not used for the classification of patients into the CTS group or non-CTS group.

Non-CTS cases were defined as cases that did not fulfill the CTS criteria described above and if there were MSKD physical signs and symptoms more prominent outside the hand distribution.

The main electrodiagnostic findings were examined as follows:

(A)Median sensory latency at Digit-II;(B)The comparative latency studies (COLSs) [6] included the following:(1)Median to ulnar latency difference through mixed nerve palmar difference study (palmdiff);(2)Median to ulnar latency difference through ring difference study (ringdiff);(3)Median to radial latency difference through thumb difference study (thumbdiff);(4)Combined sensory index (CSI), which included the sum of the latency differences of the palmdiff, ringdiff, and thumbdiff.

### 2.3. Objectives

(1)To compare the effects of age, gender, and the presence of DM on electrodiagnostic findings using linear regression.(2)To report the values of median nerve sensory latency and comparative latency studies among CTS patients and non-CTS patients. The data are presented for the whole cohort and stratified by age and presence of DM.(3)To report the best cut-off values and the diagnostic accuracy of the electrodiagnostic findings for CTS using the receiver operator characteristic (ROC) area under the curve, as well as the sensitivity, specificity, positive predictive value (PPV), and negative predictive value (NPV). We report the diagnostic accuracy in the overall cohort based on age and presence of DM, separately. The cut-offs selected in the tables presented in the manuscript are the cut-offs associated with the highest NPV associated with PPV, >90%. We chose a PPV of >90% because NCSs are typically used in clinical practice as confirmatory tests for CTS.

Data stratified by gender are in the Appendix A based on regression analysis results.

### 2.4. Data Collection

Clinical data: All patients were interviewed and examined in the clinical neurophysiology lab on the day we performed their electrodiagnostic studies as part of their clinical assessments. All data were collected prospectively. The interview was structured to inquire about the classic CTS symptoms described above and applied to all patients. Patients were examined routinely for the following MSKD and myofascial pain physical signs: (1) palpating the thumb base for tenderness for CMC arthritis; (2) Finkelstein test for de Quervain’s tenosynovitis; (3) pain on palpating the lateral epicondyle or common extensor tendon; (4) shoulder impingement test for rotator cuff syndrome (Hawkins sign); and (5) trigger points exam for fibromyalgia and myofascial pain.

Electrodiagnostic data: The nerve conduction study was performed at a warm skin temperature (>33 °C). The following electrodiagnostic tests were performed: (1) median sensory potential, stimulated at the forearm and recorded at Digit-II (14 cm); (2) ulnar sensory stimulated at the forearm and recorded at Digit V (11 cm); (3) median motor potential recorded at abductor pollicis brevis (APB) and stimulated at the wrist (7 cm) and elbow; (4) ulnar motor potential recorded at abductor digiti minimi (ADM) and stimulated at the wrist (7 cm) and above the elbow; (5) median versus ulnar antidromic sensory comparison, stimulated at the forearm and recorded at digit IV/ring finger study (ringdiff) (14 cm fixed distance for median and ulnar); (6) median versus radial sensory comparison at Digit I/thumb study (thumbdiff) (10 cm fixed distance for median and radial); and (7) median versus ulnar orthodromic mixed palmar study (palmdiff) (8 cm fixed distance for median and ulnar). We used the average of three stimulations.

### 2.5. Statistical Analysis

The demographic features were described using medians, interquartile ranges (IQRs), and frequencies. We used Mann–Whitney U tests and χ^2^ to compare the medians and proportions as appropriate. We used linear regression to compare the effects of age, gender, and the presence of DM on the electrodiagnostic findings. The diagnostic accuracy of diabetic neuropathy was evaluated using the ROC curve analysis. The diagnostic properties included sensitivity, specificity, PPV, and NPV. Statistical analyses were performed using STATA version 13 (Stata-Corp., College Station, TX, USA).

## 3. Results

Between January 2017 and July 2023, 713 cases were referred to the author’s CNPL for possible CTS. A total of 632 patients (431 of whom were female) were included in the study. Of these, 228 had no CTS and 404 had CTS. The participants’ characteristics are summarized in Table 1. The gender distribution was more males among the non-CTS group (51.7%), compared to 20.5% in the CTS group. The median age of the CTS group was higher (51 years) than that of the group without CTS (44.5 years). Patients with DM were equally distributed among both groups. The CTS group had a longer duration of symptoms and was likelier to have hypothyroidism (Table 1).

We used Bland and Padua classifications in order to grade the severity of median neuropathy (Table 1) [19,20]. These classifications used different cut-off values with different sensitivity and specificity than what we describe in this article and, when applied to our cohort, they showed similar percentages between the CTS group and non-CTS group in the severity grade of less than moderate. A recent study addressed this limitation and suggested the need to factor in demographic factors when interpreting electrodiagnostic studies, which is what our current study aimed to do [21].

### 3.1. Effect of Age, Gender, and Presence of DM on Electrodiagnostic Findings

We applied a linear regression equation, including age, gender, duration, and DM as independent variables controlled for CTS diagnosis, and the median sensory latency at Digit-II as dependent variables. The regression revealed that the coefficient was larger for the presence of DM, with a coefficient of 0.33 (95% confidence interval, CI: 0.14 to 0.53, *p* value = 0.001), followed by age groups, with a coefficient of 0.017 (95% CI: 0.011 to 0.023, *p* value < 0.000), and symptom duration, with a coefficient of 0.001 (95% CI: 0.00009 −0.003, *p* value 0.039), while the coefficient of gender was statistically insignificant at 0.1 (95% CI: −0.06 to 0.27, *p* value = 0.23). Based on these results, we present our results based on age group and presence of DM, while the data based on gender are included in the Appendix A.

In a linear regression analysis, the coefficient of an independent variable indicates the average change in the dependent variable for a one-unit change in the independent variable, with a higher coefficient value indicating a stronger effect. The coefficient for DM, 0.33, indicated that the presence of DM led to an increase in median sensory latency at Digit-II by 0.33 ms, which is consistent with DM’s impact on peripheral nerves. The coefficient for age, 0.017, showed an increase in latency by 0.017 ms per year of age. The coefficient for symptom duration, 0.001, suggested a slight increase in latency (0.001 ms) per month, which was estimated to be 0.012 ms and 0.036 ms in 50% and 75% of cases in our cohort, respectively. The coefficient for gender, 0.1, was statistically insignificant (*p* = 0.23), after controlling for DM, age, and symptom duration.

### 3.2. The Values of Median Nerve Sensory Latency and Comparative Studies among CTS Patients and Non-CTS Patients (Table 2)

For the whole cohort, the median value of median nerve sensory latency at Digit-II was 3.0–3.4 ms in the non-CTS group across all ages and increased from 3.5–3.6 ms in CTS patients aged 30–49 years to 4.1–4.4 ms in those aged over 50 years. For the palmdiff, the median value was 0.1–0.3 ms for all age groups for the non-CTS group, while, for the CTS group, it was 0.45 ms for the 30–39-year-olds and 0.7–0.9 ms for the CTS group > 40 years old. For the thumbdiff, the median value was 0.3 to 0.5 ms for all age groups for the non-CTS group, while, for the CTS group, it was 0.6 for those <30 years old, 0.8–1.0 ms for those aged 30–59 years old, and 1.4 ms for those >60 years old. For the ringdiff, the median value was 0.0 to 0.2 ms for all age groups for the non-CTS group, 0.3–0.5 ms for those <50 years old, and 0.8–1.0 ms for those >50 years.

#### 3.2.1. The Values Stratified by the Presence of DM (Table 2)

For the subgroup without DM, median nerve sensory latency at Digit-II ranged from 3.0 to 3.3 ms in the non-CTS group across all ages and increased to 3.5–3.9 ms in CTS patients aged 30–59 and 4.4 ms for those over 60 years old. For the palmdiff, the median value was 0.1 ms for the non-CTS group across all ages, except for those over 60, who recorded 0.3 ms. For CTS patients, differences increased from 0.4–0.6 ms in those aged 30–59 to 0.8 ms in those over 60 years old. For the thumbdiff, the median values were 0.3–0.4 ms in the non-CTS group across all ages, while, for the CTS group, they were 0.6 for those <30 years old, 0.8–0.9 ms for those aged 30–59 years old, and 1.1 ms for those >60 years old. For the ringdiff, the median values were 0.0–0.2 ms for the non-CTS groups across all ages, while CTS patients showed greater differences, with 0.3–0.5 ms for those under 50 and 0.7–0.8 ms for those over 50 years old.

For the subgroup with DM, the median value of the median sensory latency at Digit-II was 3 ms for the non-CTS group under 40 years old and increased to 3.4–3.6 ms for those over 40 years old. In the CTS group, the median values were 3.4–3.5 ms for those under 40 years old and rose to 4.2–4.7 ms for those over 40 years old. For the palmdiff, the median values ranged from 0.1 to 0.5 ms in the non-CTS group across all ages and were 0.5 ms in CTS patients under 40 years old and 0.9–1.6 ms in those over 40 years old. For the thumbdiff, the median values were 0.3–0.6 ms for the non-CTS group across all ages, while, for the CTS group, they increased from 0.9–1.1 ms in those under 40 years old to 1.5–1.6 ms in those over 40 years old. For the ringdiff, the median values were 0.0–0.4 ms for the non-CTS group across all ages, while, for CTS patients, they were 0.4–0.7 ms for those under 50 years old and 1.1–1.3 ms for those over 50 years old.

#### 3.2.2. The Values Stratified by Gender (Appendix A)

The median value of the median sensory latency at Digit-II was 3–3.5 ms for all age groups for the group without CTS in both males and females. For the CTS group, it was 3–3.6 ms for the age group < 50 years old and 4–4.5 ms for the age group > 50 years old for males and females. For the palmdiff, thumbdiff, ringdiff, and CSI, please refer to Appendix A.

### 3.3. Best Cut-Off Values and Diagnostic Accuracy of Carpal Tunnel Syndrome Diagnosis

#### 3.3.1. Best Cut-Off Values and Diagnostic Accuracy of Carpal Tunnel Syndrome Diagnosis in the Whole Cohort (Table 3)

For the entire cohort, the highest NPV associated with PPV over 90% was observed for the CSI, with age-specific cut-offs at 1.3 ms for ages 15–49, 2.0 ms for ages 50–59, and 3.5 ms for those over 60 years. The palmdiff followed, with cut-offs at 0.5–0.6 ms for ages 15–59 and 1.4 ms for those over 60, as detailed in Table 3. Sensitivity generally remained below 50% for median sensory latency and COLS, except for the CSI, which exceeded 50%, excluding the > 60 age group. The specificity was relatively high, mostly >90%, with the lowest being 84.3%. The optimal DA cut-offs for median sensory latency at Digit-II were between 3.7 and 4.0 ms, except for individuals over 60 years, where it was 4.7 ms. More details on specificity, PPV, and NPV for each threshold are available in the Appendix A.

#### 3.3.2. Best Cut-Off Values and Diagnostic Accuracy of Carpal Tunnel Syndrome Diagnosis Stratified by Presence of DM (Table 4 and Table 5)

For patients without DM, the highest NPV associated with PPV > 90% was the highest for the CSI, with PPV > 90% and NPV > 60% across all ages. This was followed by palmdiff. The CSI cut-offs were at 1.2–1.4 ms for ages 15–59 and 2.0 ms for those over 60 years old. For palmdiff, the cut-off was at 0.5 ms across all ages. Sensitivity averaged at 70.5% for CSI and 58.9% for palmdiff. For details about specificity and NPV at each cut-off level, see the Appendix A.

In the DM group, the highest NPV associated with PPV > 90% was the highest for the CSI, with PPV > 90% and NPV > 50% across all ages except for those over 60 years old. This was followed by thumbdiff and palmdiff. The cut-offs for CSI were 1.0–1.2 ms for ages 15–49, 3.0 ms for ages 50–59, and 3.5 ms for those over 60 years old. For thumbdiff, the cut-offs were 0.8–1.0 ms for ages 15–49, 1.2 ms for ages 50–59, and 1.8 ms for those over 60 years old. In the over-60 age group, thumbdiff and palmdiff showed the best PPV and NPV, while CSI maintained the highest sensitivity across all ages. For details about specificity and NPV at each cut-off level, see the Appendix A.

#### 3.3.3. Best Cut-Off Values and Diagnostic Accuracy of Carpal Tunnel Syndrome Diagnosis Stratified by Gender (Appendix A)

For the female group, the highest NPV associated with a PPV > 90% was the highest for the CSI, except for the age group > 60 years, which was the highest for ringdiff. For the male group, the highest NPV associated with PPV > 90% was the highest for the CSI, except for the age group > 50 years, it was the highest for median sensory latency at Digit-II. Detailed specificity, PPV, and NPV at each cut-off level are included in the Appendix A.

## 4. Discussion

Our study delineates the impact of age, gender, and diabetes on the diagnostic accuracy (DA) of comparative latency studies (COLSs) for carpal tunnel syndrome (CTS). By stratifying data according to these variables, we revealed significant variations that have important implications for clinical practice. The most important finding in our study is that we reported the cut-off values that produce the best DA according to different age groups in the presence and absence of diabetes. For PPV > 90% and NPV > 50%, the best cut-offs among patients without DM were 0.5 ms, 0.8–1 ms, 0.6 ms, 1.4 ms, and 2 ms for palmdiff, thumbdiff, ringdiff, CSI (age < 60 years), and CSI (age > 60 years), respectively. The best cut-offs among patients with DM were 0.5 ms, 1.2 ms, 0.8 ms, 1.0–1.2 ms, 1.8 ms, 0.5 ms, 1.7 ms, 1–1.2 ms, 3.0 ms, and 3.5 ms for palmdiff (age < 50 years), palmdiff (age > 50 years), thumbdiff (age < 40 years), thumbdiff (age 40–59 years), thumbdiff (age > 60 years), ringdiff (age < 50 years), ringdiff (age > 50), CSI (age < 50 years), CSI (age 50–59 years), and CSI (age > 60 years), respectively. Notably, our findings for overall sensitivity and specificity are somewhat lower than previously reported figures. Robinson et al. reported overall sensitivity of 69.7%, 74.2%, 75.8%, and 83.1% and specificity of 96.9%, 96.9%, 96.9%, and 95.4% for palmdiff, ringdiff, thumbdiff, and CSI, respectively [10]. They also reported PPV of 95.8%, 96.2%, 96.2%, and 94.8% and NPV of 76.8%, 78.8%, 79.7%, and 85.0% for palmdiff, ringdiff, thumbdiff, and CSI, respectively [10]. In comparison, we report sensitivity of 57.7%, 39.8%, 45.9%, and 60.3% and specificity of 91.9%, 94.7%, 93%, and 89.5% for palmdiff, ringdiff, thumbdiff, and CSI, respectively. We report PPV of 91.5%, 91.3%, 90.7%, and 90% and NPV of 58.9%, 53.2%, 53.6%, and 59.1% for palmdiff, ringdiff, thumbdiff, and CSI, respectively. Other studies have reported similar DA values to Robinson et al [3]. Three reasons may account for the difference between our findings and those in the literature. First, the cut-off values we selected aimed to ensure a PPV of over 90%, leading to variations in sensitivity and specificity when compared to studies that used different cut-offs (e.g., 0.4 ms for palmdiff, 0.5 ms for thumbdiff, and 1.0 ms for CSI). Although aligning our cut-off values with those used in previous studies could increase our sensitivity to the levels reported in the literature, this would potentially compromise specificity and PPV. Second, our study cohort included a more diverse population, as it encompassed all patients referred to our clinical neurophysiology lab (CNPL) for potential CTS, using non-CTS cases (including those with musculoskeletal diseases and other neuropathies, such as ulnar neuropathy and cervical radiculopathy) as controls. This approach likely provided a more accurate reflection of the disease’s prevalence in clinical settings compared to studies that used preselected control groups without symptomatic hand issues. Third, we included patients with DM in our cohort, while it was not reported whether patients with DM were included in prior studies, such as that by Robinson et al. When we excluded patients with DM (Table 4), the DA components became closer to the reported DA in the literature.

Our data highlight the significant impact of DM on the electrodiagnostic parameters used to diagnose CTS. Notably, the cut-off values for diagnosing CTS are less variable by age in patients without DM. The palmdiff cut-off values to reach >90% PPV were 0.5 ms across all ages, and the cut-off values were 0.8–1.0 ms, 0.6–0.7 ms, and 1.2–2.0 ms across all ages for thumbdiff, ringdiff, and CSI, respectively. In contrast, the cut-off value to reach >90% PPV for CTS in patients with DM showed wide variation based on age group, and the values ranged between 0.5 and 1.4 ms, 0.8 and 1.8 ms, 0.5 and 1.7 ms, and 1.0 and 3.5 ms across different age groups in palmdiff, thumbdiff, ringdiff, and CSI, respectively. Albers et al. suggested that patients with diabetes should have their own diagnostic criteria for CTS, as the median latency is expected to be prolonged without clinical carpal tunnel syndrome [22]. Albers et al. found that sural and superficial peroneal responses did not influence median nerve electrodiagnostic findings. It has been estimated that 23–27% of patients with diabetes exhibit electrodiagnostic features of median neuropathy at the wrist without having clinical features of CTS [22,23]. Diabetes is considered a risk factor for developing CTS [24]. Nonetheless, diabetes does not mitigate the benefits of CTS decompression surgeries, which are shown to prevent hand weakness effectively if the diagnosis is made before significant axonal loss occurs [7]. Despite the fact that ultrasounds potentially help to diagnose CTS, some studies found that electrodiagnostics are more accurate than ultrasound for CTS diagnosis in DM patients [13,25,26]. This was attributed to the possibility that the ultrasound cross-sectional area (CSA) is not increased in chronic or severe CTS, which is likelier to occur with DM and CTS; however, this was based on indirect evidence of a lack of correlation between ultrasound findings (e.g., CSA) and electrodiagnostic findings among CTS patients with DM and among moderate to severe CTS cases [25,27]. More direct evidence is needed to prove or dispute this theory. We established cut-off values based on age group in the presence and absence of DM, which helped to diagnose CTS in patients with DM.

Age is an important risk factor for CTS [2]. We reported the DA as per age group. In the absence of DM, we found that the cut-off values were not influenced by age for palmdiff or ringdiff, but they affected thumbdiff and CSI, with a higher cut-off value for those older than 60 years. However, in the presence of DM, the cut-off values were largely affected. The importance of specific cut-off values for COLS based on age was emphasized, as the aging effects on the median nerve are greater than its effect on the ulnar nerve [28]. Werner et al. found that the conduction velocity of the median sensory nerve decreased by 0.41 m/s per year, while that of the ulnar sensory nerve decreased by 0.29 m/s per year. Another study found that sensory latency was prolonged by 0.11 ms for the median nerve and 0.06 ms for the ulnar nerve over a five-year period [11]. Despite various studies on how age affects outcomes following CTS surgery, no prior research has established age-specific DA for COLS. Some studies have shown that elderly patients are less satisfied with CTS release surgery [29]. One reason is that regeneration is slower as age increases [29,30]. Other studies found no influence of age on the outcome of CTS release, and this was attributed to the exclusion of patients with DM and thyroid disease [31]. The lack of age-related effects on surgical results in studies excluding DM cases supports our findings that age significantly impacts DA among diabetic patients, underscoring the necessity for more precisely tailored cut-off values to accurately diagnose CTS in these individuals.

Our study has several limitations. First, our analysis of diagnostic accuracy (DA) and cut-off values was limited to combinations of only two factors—age with the presence of DM and age with gender—due to an insufficient sample size to integrate all three factors together. Second is the retrospective nature of the study; however, data collection was conducted prospectively through structured interviews, which enhanced the reliability of our findings. Third, we defined cases of CTS using clinical criteria, which, despite some limitations, is a widely accepted approach among researchers. However, it may not capture all CTS cases and we had to exclude some cases to minimize misclassification bias. Fourth, our analysis did not consider additional variables that might affect sensory latency, such as occupational hand strain. Fifth, we included cases performed by a single neurophysiologist, and the reason for this is that the other co-workers did not perform all three COLSs in all patients, and the structured interview and musculoskeletal exam were not standard procedure in their CNPL. Sixth, we excluded some cases that could not be classified into CTS or non-CTS cases in order to avoid misclassification bias. Excluding these borderline cases may have yielded a selection bias that may have affected the generalizability of this study. Selection bias is crucial to avoid; however, we opted to minimize misclassification bias over selection bias as selection bias impact can be somewhat mitigated through careful interpretation and acknowledgment of the limitations. Despite these limitations, we believe that our data are informative for clinical practice, as we stratified the data based on age and the presence of DM and included all CNPL-referred patients, which makes our DA data more applicable clinically.

## 5. Conclusions

In conclusion, our study demonstrated that electrodiagnostic studies can be used to confirm CTS diagnosis with a PPV > 90%. Additionally, our study provided specific cut-off values, with DA based on age and DM, which can guide the clinical practice, ensuring that diagnostic strategies are both precise and tailored to individual patient profiles, particularly in the context of aging and diabetes. We used clinical criteria to classify patients into CTS or non-CTS groups, which may not have captured all CTS cases. Additionally, we excluded cases that we could not classify into either group to avoid misclassification. These factors may have limited the generalizability of our conclusions. Nonetheless, our data are an important step toward a more accurate diagnosis of CTS.

## Figures and Tables

**Table 1 diagnostics-14-01381-t001:** Participants’ characteristics.

	No Carpal Tunnel Syndrome (*n* = 228)	Carpal Tunnel Syndrome (*n* = 404)	*p* Value
Age, median (IQR)	44.5 (36–56.5)	51 (42–58)	0.0001
Male, n (%)	118 (51.75)	83 (20.54)	0.000
DM, n (%)	49 (21.49)	105 (25.99)	0.212
Hypothyroidism, n (%) (n = 594)	20 (9.85)	71 (18.16)	0.008
Duration of symptoms, months, median (IQR)	7.5 (2–36)	12 (5–36)	0.0014
Right hand dominant, n (%) (n = 421)	142 (92.81)	247 (92.16)	0.69
Number of cases per age group			
AG (<30 years), n (%)	34 (14.9)	23 (5.69)	0.000
AG (30–39 years), n (%)	44 (19.3)	54 (13.37)	
AG (40–49 years), n (%)	55 (24.12)	96 (23.76)	
AG (50–59 years), n (%)	51 (22.37)	150 (37.13)	
AG (>60 years), n (%)	44 (19.3)	81 (20.05)	
Diagnosis of non-CTS cases			
Ulnar neuropathy, n (%)	55 (24.1)	0	
Cervical radiculopathy, n (%)	30 (13.2)	0	
Musculoskeletal and other causes, n (%)	143 (62.7)	0	
Severity classification of median neuropathy			
* Bland classification			
0: normal, n (%)	146 (64.0)	113 (27.9)	0.00
1: very mild, n (%)	29 (12.7)	52 (12.8)	
2: mild, n (%)	42 (18.4)	91 (22.5)	
3: moderate, n (%)	10 (4.3)	99 (24.5)	
4: severe, n (%)	1 (0.44)	17 (4.2)	
5: very severe, n (%)	0	28 (6.9)	
6: extremely severe, n (%)	0	4 (0.99)	
* Padua classification			
0: normal, n (%)	70 (30.7)	35 (8.6)	0.00
1: minimal, n (%)	25 (10.9)	22 (5.4)	
2: mild, n (%)	105 (46.0)	109 (26.9)	
3: moderate, n (%)	27 (11.8)	188 (46.5)	
4: severe, n (%)	1 (0.44)	46 (11.39)	
5: extreme, n (%)	0	4 (0.99)	

IQR: interquartile range. AG: age group. * Bland and Padua classification used different cut-off values with different sensitivity and specificity than what we describe in this article.

**Table 2 diagnostics-14-01381-t002:** Values of median nerve latencies and comparative latency studies (COLSs).

	Whole Cohort			No DM			DM		
	No CTS ms Median (IQR)/95% ULN	CTS ms Median (IQR)	*p* Value	No CTS ms Median (IQR)/95% ULN	CTS ms Median (IQR)	*p* Value	No CTS ms Median (IQR)/95% ULN	CTS ms Median (IQR)	*p* Value
Median sensory latency at Digit-II									
All age groups	3.2 (3–3.4)/4.2	3.8 (3.3–4.8)	0.000	3.1 (2.9–3.3)/3.8	3.7 (3.3–4.5)	0.000	3.5 (3.2–4)/4.8	4.3 (3.7–5.5)	0.000
Group 1 < 30 years	3 (2.8–3.2)/4.3	3 (2.8–3.5)	0.52	3 (2.8–3.2)/3.7	3 (2.8–3.5)	0.734	2.9 (2.9–2.9)/2.9	3.5 (3–4)	0.22
Group 2 30–39 years	3.1 (2.9–3.2)/3.6	3.5 (3.1–4.4)	0.0001	3.1 (2.9–3.2)/3.6	3.5 (3.1–4.4)	0.000	3 (3–3)/3	3.4 (2.8–4.1)	0.654
Group 3 40–49 years	3 (2.9–3.3)/3.9	3.6 (3.3–4.5)	0.000	3 (2.9–3.3)/3.9	3.6 (3.3–4.3)	0.000	3.6 (2.9–3.7)/4	4.2 (3.3–5.3)	0.156
Group 4 50–59 years	3.3 (3.1–3.6)/4.3	4.1 (3.5–5.3)	0.000	3.2 (3.1–3.4)/3.6	3.9 (3.4–5.2)	0.000	3.4 3.3–3.9)/4.9	4.7 (3.7–5.7)	0.001
Group 5 > 60 years	3.4 (3.2–3.8)/4.6	4.4 (3.7–5.2)	0.000	3.3 (3.1–3.6)/3.8	4.4 (3.7–4.8)	0.000	3.6 (3.3–4.2)/4.8	4.4 (3.9–5.7)	0.000
Median motor latency at abductor pollicis brevis (APB)									
All age groups	3.3 (3.0–3.6)/4.8	4.2 (3.5–5.7)	0.000	3.2 (3–3.5)/4	4 (3.4–5.2)	0.000	3.8 (3.3–4.1)/5.4	5.3 (4.2–6.5)	0.000
Group 1 < 30 years	3.1 (3–3.3)3.5	3.1 (2.8–3.9)	0.64	3.1 (3–3.3)/3.5	3.1 (2.8–3.8)	0.936	2.8 (2.8–2.8/2.8	3.5 (3.2–3.9)	0.22
Group 2 30–39 years	3.2 (3–3.4)/4	3.7 (3.2–4.5)	0.000	3.2 (93–3.4)/4	3.7 (3.2–4.5)	0.000	3.4 (3.4–3.4)/3.4	3.7 (3.1–6.0)	1.00
Group 3 40–49 years	3.3 (3.1–3.6)/4	4 (3.4–5.4)	0.000	3.2 (3.1–3.5)/4	3.9 (3.4–4.9)	0.000	3.6 (3.2–3.8)/3.9	5.5 (4.3–6.4)	0.15
Group 4 50–59 years	3.3 (3.1–3.8)/5.3	4.4 (3.6–5.9)	0.000	3.3 (3–3.6)/3.9	4.1 (3.5–5.9)	0.000	3.7 (3.2–4.2)/6.1	5.2 (4.2–6.3)	0.000
Group 5 > 60 years	3.8 (3.3–4.1)/4.9	5.2 (4.3–6.5)	0.000	3.5 (3.2–4)/4.8	5.1 (4.2–6.1)	0.000	3.9 (3.4–4.2)/5.1	5.8 (4.3–6.7)	0.000
Mixed palmar studies (palmdiff): median latency (palm)–ulnar latency(palm)									
All age groups	0.1 (0–0.3)/0.8	0.7 (0.3–1.3)	0.000	0.1 (0–0.3)/0.5	0.6 (0.3–1.1)	0.000	0.3 (0.1–0.7)/1.3	1 (0.4–1.6)	0.000
Group 1 < 30 years	0.1 (0–0.2)/0.3	0.25 (0–0.4)	0.036	0.1 (0–0.2)/0.3	0.1 (0–0.3)	0.091	0.1 (0.1–0.1)/0.1	0.5 (0.3–0.7)	0.22
Group 2 30–39 years	0.1 (0–0.3)/0.4	0.45 (0.2–1)	0.000	0.1 (0–0.3)/0.4	0.4 (0.2–1)	0.000	0 (0–0)/0	0.5 (0.2–0.6)	0.179
Group 3 40–49 years	0.1 (0–0.2)/0.6	0.8 (0.3–1.3)	0.000	0.1 (0–0.2)/0.6	0.6 (0.3–1.1)	0.000	0.2 (0.15–0.2)/0.3	1.6 (0.7–2.3)	0.006
Group 4 50–59 years	0.2 (0–0.4)/0.8	0.7 (0.4–1.4)	0.000	0.1 (0–0.3)/0.5	0.6 (0.4–1.3)	0.000	0.5 (0.04–0.8)/1.6	0.9 (0.4–1.6)	0.15
Group 5 > 60 years	0.3 (0.1–0.4)/1.2	0.9 (0.6–1.5)	0.000	0.3 (0–0.3)/0.5	0.8 (0.6–1.7)	0.000	0.3 (0.2–0.7)/1.3	1 (0.4–1.5)	0.001
Median to radial comparison, thumbdiff: median latency (D1)–radial latency (D1)									
All age groups	0.4 (0.2–0.6)/1.2	1.0 (0.6–1.7)	0.000	0.3 (0.2–0.5)/1	0.9 (0.6–1.6)	0.000	0.6 (0.3–1)/1.7	1.5 (0.9–2.3)	0.000
Group 1 < 30 years	0.3 (0.2–0.4)/0.6	0.6 (0.4–1.1)	0.0001	0.3 (0.2–0.4)/0.6	0.6 (0.3–0.8)	0.000	0 (0–0)/0	1.1 (1.1–1.2)	0.22
Group 2 30–39 years	0.35 (0.1–0.6)/1	0.8 (0.4–1.2)	0.0002	0.4 (0.1–0.6)/1	0.8 (0.4–1.2)	0.000	0.3 (0.3–0.3)/0.3	0.9 (0.2–1.4)	0.654
Group 3 40–49 years	0.3 (0.2–0.6)/1.1	1 (0.6–1.6)	0.000	0.3 (0.2–0.5)/1.1	0.9 (0.5–1.4)	0.000	0.6 (0.6–0.9)/1.1	1.6 (0–2.4)	0.38
Group 4 50–59 years	0.4 (0.3–0.6)/1.1	1 (0.7–1.8)	0.000	0.4 (0.4–0.5)/0.9	0.9 (0.6–1.7)	0.000	0.5 (0.4–0.9)/2	1.5 (0.9–2.2)	0.000
Group 5 > 60 years	0.5 (0.3–0.9)/1.4	1.4 (0.8–2.3)	0.000	0.4 (0.3–0.8)/1.4	1.1 (0.9–1.9)	0.000	0.6 (0.3–1.0)/1.7	1.5 (0.6–2.9)	0.001
Median to ulnar comparison at ring finger study (ringdiff): Median latency (Digit IV)-Ulnar latency(Digit IV)									
All age groups	0.1 (0–0.3)/1	0.7 (0.3–1.5)	0.000	0.1 (0–0.3)/0.6	0.6 (0.2–1.4)	0.000	0.3 (0.04–1)/1.6	1 (0.4–1.9)	0.000
Group 1 < 30 years	0 (0–0.02)/0.4	0.3 (0.1–0.5)	0.0036	0 (0–0.2)/0.4	0.3 (0.1–0.5)	0.004	0.4 (0.4–0.4)/0.4	0.7 (0.1–1.4)	1.00
Group 2 30–39 years	0.1 (0–0.2)/0.5	0.4 (0.1–1.2)	0.003	0.1 (0–0.2)/05	0.4 (0.1–1.3)	0.000	0 (0–0)/0	0.4 (0–0.6)	0.345
Group 3 40–49 years	0 (0–0.2)/0.7	0.5 (0.3–1.4)	0.000	0 (0–0.2)/0.7	0.5 (0.3–1.4)	0.000	0.2 (0.2–0.3)/0.3	0.6 (0.3–2.4)	0.217
Group 4 50–59 years	0.1 (0–0.5)/1.4	0.8 (0.35–1.8)	0.000	0.1 (0–0.4)/0.8	0.7 (0.3–1.7)	0.000	0.3 (0–0.9)/3.4	1.3 (0.5–2.4)	0.02
Group 5 > 60 years	0.2 (0–0.6)/1.4	1 (0.6–1.4)	0.000	0.2 (0–0.4)/0.6	0.8 (0.6–1.3)	0.000	0.35 (0.1–1.1)/1.4	1.1 (0.6–1.5)	0.019
Combined sensory index: sum of palmdiff + thumbdiff + ringdiff									
All age groups	0.6 (0.4–1.1)/2.6	2.4 (1.3–4.25)	0.000	0.6 (0.3–0.9)/1.9	2.2 (1.2–3.9)	0.000	1.1 (0.6–2.6)/4.2	3.5 (2.3–5.4)	0.000
Group 1 < 30 years	0.5 (0.3–0.7)/1.1	1.1 (0.6–1.6)	0.000	0.55 (0.3–0.7)/1.1	1 (0.6–1.6)	0.001	0.5 (0.5–0.5)/0.5	2.3 (1.5–3.2)	0.22
Group 2 30–39 years	0.5 (0.3–0.9)/1.4	1.6 (0.5–3.3)	0.000	0.55 (0.3–0.9)/1.4	1.5 (0.5–3.3)	0.000	0.3 (0.3–0.3)/0.3	1.8 (0.4–2.4)	0.179
Group 3 40–49 years	0.6 (0.2–0.8)/2.3	2.3 (1.3–4.4)	0.000	0.5 (0.2–0.8)/2.3	2.25 (1.1–4.2)	0.000	1 (0.85–1.1)/1.2	4.6 (2.3–5.6)	0.01
Group 4 50–59 years	0.7 (0.4–1.4)/2.6	2.6 (1.6–4.6)	0.000	0.6 (0.4–1)/2	2.3 (1.4–4.1)	0.000	1.3 (0.6–2.6)/5	3.8 (2.5–6.2)	0.002
Group 5 > 60 years	1.1 (0.6–1.9)/4.2	3.2 (2.3–4.7)	0.000	0.7 (0.4–1.8)/1.9	3.1 (2.3–4.4)	0.000	1.4 (0.6–2.8)/4	3.5 (2.3–4.9)	0.001

IQR: interquartile range. ULN: upper limit of normal. ms: milliseconds.

**Table 3 diagnostics-14-01381-t003:** Cut-off values and diagnostic accuracy of median nerve and comparative latency studies (COLSs), whole cohort.

Age Group	Cut-Off (ms)	ROC	Sensitivity	Specificity	PPV	NPV
	Median sensory latency at Digit-II					
Whole cohort	4.1 ms	0.685 (0.655–0.715)	43.2% (37.9–48.5%)	93.8% (89.9–96.6%)	91.6% (86.3–95.3%)	51.6% (46.6–56.5%)
Group 1 < 30 years	4.1 ms	0.543 (0.485–0.602)	8.7% (1.07–28%)	100% (89.7–100%)	100% (15.8–100%)	61.8% (47.7–74.6%)
Group 2 30–39 years	3.7 ms	0.702 (0.626–0.778)	45.1% (31.1–59.7%)	95.3% (84.2–99.4%)	92% (74–99%)	59.4% (46.9–71.1%)
Group 3 40–49 years	4.0 ms	0.678 (0.62–0.736)	39.3% (28.8–50.5%)	96.4% (87.5–99.6%)	94.3% (80.8–99.3%)	51% (41–60.9%)
Group 4 50–59 years	3.7 ms	0.737 (0.672–0.801)	63% (54.4–71.1%)	84.3% (71.4–93%)	91.6% (84.1–96.3%)	45.7% (35.4–56.3%)
Group 4 > 60 years	4.7 ms	0.665 (0.594–0.736)	37.5% (24.9–51.5%)	95.5% (84.5–99.4%)	91.3% (72–98.9%)	54.5% (42.8–65.9%)
Palmdiff	Mixed palmar studies (palmdiff): median latency (palm)–ulnar latency (palm)					
Whole cohort	0.6	0.748 (0.715–0.781)	57.7% (52.1–63.2%)	91.9% (87.3–95.2%)	91.5% (86.7–95%)	58.9% (53.3–64.3%)
Group 1 < 30 years	0.5	0.614 (0.524–0.703)	22.7% (7.82–45.4%)	100% (89.7–100%)	100% (47.8–100%)	66.7% (52.1–79.2%)
Group 2 30–39 years	0.5	0.737 (0.662–0.813)	50% (35.2–64.8%)	97.5% (86.8–99.9%)	96% (79.6–99.9%)	61.9% (48.8–73.9%)
Group 3 40–49 years	0.4	0.796 (0.727–0.864)	71.6% (60.5–81.1%)	87.5% (74.8–95.3%)	90.6% (80.7–96.5%)	64.6% (51.8–76.1%)
Group 4 50–59 years	0.6	0.727 (0.659–0.795)	60.3% (50.8–69.3%)	85.1% (71.7–93.8%)	90.9% (82.2–96.3%)	46.5% (35.7–57.6%)
Group 4 > 60 years	1.4	0.647 (0.578–0.717)	32% (19.5–46.7%)	97.5% (86.8–99.9%)	94.1% (71.3–99.9%)	53.4% (41.4–65.2%)
Thumdiff	Median to radial comparison, thumbdiff: median latency (D1)–radial latency (D1)					
Whole cohort	1.1	0.695 (0.663–0.727)	45.9% (40.4–51.6%)	93% (88.8–96%)	90.7% (85.2–94.7%)	53.6% (48.4–58.8%)
Group 1 < 30 years	0.7	0.688 (0.579–0.798)	40.9% (20.7–63.6%)	96.8% (83.3–99.9%)	90% (55.5–99.7%)	69.8% (53.9–82.8%)
Group 2 30–39 years	1.1	0.658 (0.587–0.728)	34% (21.2–48.8%)	97.5% (86.8–99.9%)	94.4% (72.7–99.9%)	54.2% (42–66%)
Group 3 40–49 years	1.0	0.715 (0.65–0.781)	50.6% (39.3–61.9%)	92.5% (81.8–97.9%)	91.1% (78.8–97.5%)	55.1% (44.1–65.6%)
Group 4 50–59 years	1.0	0.724 (0.664–0.784)	53% (43.5–62.4%)	91.8% (80.4–97.7%)	93.8% (85–98.3%)	45.5% (35.4–55.8%)
Group 4 > 60 years	1.5	0.707 (0.631–0.783)	46.2% (32.2% 60.5%)	95.2% (83.8–99.4%)	92.3% (74.9–99.1%)	58.8% (46.2–70.6%)
Ringdiff	Median to ulnar comparison at ring finger study (ringdiff): median latency (Digit IV)–ulnar latency (Digit IV)					
Whole cohort	1.0	0.673 (0.641–0.705)	39.8% (34.1–45.7%)	94.7% (90.8–97.3%)	91.3% (84.9–95.6%)	53.2% (48–58.4%)
Group 1 < 30 years	0.7	0.614 (0.524–0.703)	22.7% (7.82–45.4%)	100% (89.1–100%)	100% (47.8–100%)	65.3% (50.4–78.3%)
Group 2 30–39 years	0.6	0.71 (0.633–0.787)	44.4% (29.6–60%)	97.6% (87.1–99.9%)	95.2% (76.2–99.9%)	61.5% (48.6–73.3%)
Group 3 40–49 years	0.5	0.769 (0.701–0.837)	62.2% (50.1–73.2%)	91.7% (80–97.7%)	92% (80.8–97.8%)	61.1% (48.9–72.4%)
Group 4 50–59 years	1.0	0.686 (0.625–0.747)	45.4% (35.8–55.2%)	91.8% (80.4–97.7%)	92.5% (81.8–97.9%)	43.3% (33.6–53.3%)
Group 4 > 60 years	1.3	0.6 (0.529–0.67)	22.5% (10.8–38.5%)	97.4% (86.5–99.9%)	90% (55.5–99.7%)	55.1% (42.6–67.1%)
CSI	Combined sensory index: sum of palmdiff + thumbdiff + ringdiff					
Whole cohort	2.0	0.749 (0.714–0.783)	60.3% (54.6–65.7%)	89.5% (84.4–93.4%)	90% (85.1–93.7%)	59.1% (53.3–64.7%)
Group 1 < 30 years	1.2	0.734 (0.622–0.845)	50% (28.2–71.8%)	96.8% (83.3–99.9%)	91.7% (61.5–99.8%)	73.2% (57.1–85.8%)
Group 2 30–39 years	1.3	0.759 (0.678–0.841)	60% (45.2–73.6%)	91.9% (78.1–98.3%)	90.9% (75.7–98.1%)	63% (48.7–75.7%)
Group 3 40–49 years	1.3	0.821 (0.753–0.888)	76.9% (66–85.7%)	87.2% (74.3–95.2%)	90.9% (81.3–96.6%)	69.5% (56.1–80.8%)
Group 4 50–59 years	2.0	0.754 (0.686–0.822)	66.1% (56.7–74.7%)	84.8% (71.1–93.7%)	91.6% (83.4–96.5%)	50% (38.5–61.5%)
Group 4 > 60 years	3.5	0.685 (0.602–0.768)	44.7% (30.2–59.9%)	92.3% (79.1–98.4%)	87.5% (67.6–97.3%)	58.1% (44.8–70.5%)

Ms: milliseconds. ROC: receiver operator characteristic curve.

**Table 4 diagnostics-14-01381-t004:** Cut-off values and diagnostic accuracy of median nerve and comparative latency studies (COLSs) (participants without DM).

Age Group	Cut-Off (ms)	ROC	Sensitivity	Specificity	PPV	NPV
	Median sensory latency at Digit-II					
Whole cohort	3.7	0.727 (0.692–0.762)	52.2% (46.1–58.2%)	93.3% (88.5–96.5%)	92.3% (86.9–95.9%)	55.9% (50–61.6%)
Group 1 < 30 years	4.1	0.548 (0.483–0.612)	9.52% (1.17–30.4%)	100% (89.4–100%)	100% (15.8–100%)	63.5% (49–76.4%)
Group 2 30–39 years	3.7	0.705 (0.627–0.784)	45.8% (31.4–60.8%)	95.2% (83.8–99.4%)	91.7% (73–99%)	60.6% (47.8–72.4%)
Group 3 40–49 years	4.0	0.675 (0.616–0.734)	37% (26–49.1%)	98% (89.4–99.9%)	96.4% (81.7–99.9%)	51.6% (41.1–62%)
Group 4 50–59 years	3.5	0.742 (0.657–0.826)	71.8% (62.1–80.3%)	76.5% (58.8–89.3%)	90.2% (81.7–95.7%)	47.3% (33.7–61.2%)
Group 4 > 60 years	3.7	0.844 (0.741–0.947)	79.3% (60.3–92%)	89.5% (66.9–98.7%)	92% (74–99%)	73.9% (51.6–89.8%)
Palmdiff	Mixed palmar studies (palmdiff): median latency (palm)–ulnar latency (palm)					
Whole cohort	0.5	0.768 (0.733–0.803)	58.9% (52.5–65.2%)	94.6% (90–97.5%)	94.2% (89.2–97.3%)	61% (54.8–67%)
Group 1 < 30 years	0.5	0.6 (0.51–0.69)	20% (5.73–43.7%)	100% (89.4–100%)	100% (39.8–100%)	67.3% (52.5–80.1%)
Group 2 30–39 years	0.5	0.732 (0.654–0.81)	48.9% (33.7–64.2%)	97.4% (86.5–99.9%)	95.7% (78.1–99.9%)	62.3% (49–74.4%)
Group 3 40–49 years	0.5	0.753 (0.68–0.826)	59.7% (47–71.5%)	90.9% (78.3–97.5%)	90.9% (78.3–97.5%)	59.7% (47–71.5%)
Group 4 50–59 years	0.5	0.778 (0.708–0.848)	64.4% (53.4–74.4%)	91.2% (76.3–98.1%)	94.9% (85.9–98.9%)	50% (37–63%)
Group 4 > 60 years	0.5	0.897 (0.807–0.986)	85.2% (66.3–95.8%)	94.1% (71.3–99.9%)	95.8% (78.9–99.9%)	80% (56.3–94.3%)
Thumdiff	Median to radial comparison, thumbdiff: median latency (D1)–radial latency (D1)					
Whole cohort	1.0	0.706 (0.671–0.742)	46.6% (40.3–53%)	94.7% (90.1–97.5%)	92.9% (86.9–96.7%)	54.4% (48.5–60.2%)
Group 1 < 30 years	0.8	0.65 (0.547–0.753)	30% (11.9–54.3%)	100% (88.4–100%)	100% (54.1–100%)	68.2% (52.4–81.4%)
Group 2 30–39 years	1.1	0.657 (0.584–0.73)	34% (20.9–49.3%)	97.4% (86.5–99.9%)	94.1% (71.3–99.9%)	55.1% (42.6–67.1%)
Group 3 40–49 years	0.8	0.752 (0.682–0.823)	58.8% (46.2–70.6%)	91.7% (80–97.7%)	90.9% (78.3–97.5%)	61.1% (48.9–72.4%)
Group 4 50–59 years	0.9	0.752 (0.682–0.823)	58.8% (46.2–70.6%)	91.7% (80–97.7%)	90.9% (78.3–97.5%)	61.1% (48.9–72.4%)
Group 4 > 60 years	1.0	0.789 (0.676–0.903)	69% (49.2–84.7%)	88.9% (65.3–98.6%)	90.9% (70.8–98.9%)	64% (42.5–82%)
Ringdiff	Median to ulnar comparison at ring finger study (ringdiff): median latency (Digit IV)–ulnar latency (Digit IV)					
Whole cohort	0.6	0.725 (0.689–0.761)	50.4% (43.8–57%)	94.6% (90–97.5%)	92.9% (87–96.7%)	57.5% (51.4–63.4%)
Group 1 < 30 years	0.7	0.6 (0.51–0.69)	20% (5.73–43.7%)	100% (88.8–100%)	100% (39.8–100%)	66% (50.7–79.1%)
Group 2 30–39 years	0.6	0.714 (0.634–0.794)	45.2% (29.8–61.3%)	97.5% (86.8–99.9%)	95% (75.1–99.9%)	62.9% (49.7–74.8%)
Group 3 40–49 years	0.6	0.702 (0.631–0.772)	47% (34.6–59.7%)	93.3% (81.7–98.6%)	91.2% (76.3–98.1%)	54.5% (42.8–65.9%)
Group 4 50–59 years	0.6	0.729 (0.655–0.803)	54.9% (43.5–65.9%)	90.9% (75.7–98.1%)	93.8% (82.8–98.7%)	44.8% (32.6–57.4%)
Group 4 > 60 years	0.6	0.866 (0.765–0.967)	79.2% (57.8–92.9%)	94.1% (71.3–99.9%)	95% (75.1–99.9%)	76.2% (52.8–91.8%)
CSI	Combined sensory index: sum of palmdiff + thumbdiff + ringdiff					
Whole cohort	1.4	0.792 (0.753–0.83)	70.4% (64.3–76.1%)	87.9% (81.7–92.6%)	90.2% (85.1–94%)	65.4% (58.6–71.8%)
Group 1 < 30 years	1.2	0.708 (0.592–0.825)	45% (23.1–68.5%)	96.7% (82.8–99.9%)	90% (55.5–99.7%)	72.5% (56.1–85.4%)
Group 2 30–39 years	1.3	0.756 (0.672–0.841)	59.6% (44.3–73.6%)	91.7% (77.5–98.2%)	90.3% (74.2–98%)	63.5% (49–76.4%)
Group 3 40–49 years	1.4	0.79 (0.716–0.864)	69.7% (57.1–80.4%)	88.4% (74.9–96.1%)	90.2% (78.6–96.7%)	65.5% (51.9–77.5%)
Group 4 50–59 years	1.1	0.806 (0.723–0.889)	85.4% (76.3–92%)	75.8% (57.7–88.9%)	90.5% (82.1–95.8%)	65.8% (48.6–80.4%)
Group 4 > 60 years	2.0	0.92 (0.847–0.993)	84% (63.9–95.5%)	100% (78.2–100%)	100% (83.9–100%)	78.9% (54.4–93.9%)

Ms: milliseconds. ROC: receiver operator characteristic curve.

**Table 5 diagnostics-14-01381-t005:** Cut-off values and diagnostic accuracy of median nerve and comparative latency studies (COLSs) (participants with DM).

Age Group	Cut-Off (ms)	ROC	Sensitivity	Specificity	PPV	NPV
	Median sensory latency at Digit-II					
Whole cohort	4.5	0.68 (0.61–0.75)	46.2% (34.8–57.8%)	89.8% (77.8–96.6%)	87.8% (73.8–95.9%)	51.2% (40.1–62.1%)
Group 1 < 30 years	3.5	0.75 (0.0–1)	50% (1.26–98.7%)	100% (2.5–100%)	100% (2.5–100%)	50% (1.26–98.7%)
Group 2 30–39 years	3.5	0.667 (0.0–1)	33.3% (0.84–90.6%)	100% (2.5–100%)	100% (2.5–100%)	33.3% (0.84–90.6%)
Group 3 40–49 years	4.4	0.727 (0.573–0.882)	45.5% (16.7–76.6%)	100% (47.8–100%)	100% (47.8–100%)	45.5% (16.7–76.6%)
Group 4 50–59 years	4.4	0.727 (0.612–0.842)	57.1% (39.4–73.7%)	88.2% (63.6–98.5%)	90.9% (70.8–98.9%)	50% (31.3–68.7%)
Group 4 > 60 years	4.5	0.681 (0.565–0.797)	48.1% (28.7–68.1%)	88% (68.8–97.5%)	81.3% (54.4–96%)	61.1% (43.5–76.9%)
Palmdiff	Mixed palmar studies (palmdiff): median latency (palm)–ulnar latency (palm)					
Whole cohort	1.2	0.69 (0.619–0.76)	45.1% (33.2–57.3%)	92.9% (80.5–98.5%)	91.4% (76.9–98.2%)	50% (38.5–61.5%)
Group 1 < 30 years	0.5	0.75 (0.0–1)	50% (1.26–98.7%)	100% (2.5–100%)	100% (2.5–100%)	50% (1.26–98.7%)
Group 2 30–39 years	0.5	0.833 (0.0–1)	66.7% (9.43–99.2%)	100% (2.5–100%)	100% (15.8–100%)	50% (1.26–98.7%)
Group 3 40–49 years	0.5	0.929 (0.833–1)	85.7% (57.2–98.2%)	100% (39.8–100%)	100% (73.5–100%)	66.7% (22.3–95.7%)
Group 4 50–59 years	1.2	0.703 (0.584–0.822)	48.3% (29.4–67.5%)	92.3% (64–99.8%)	93.3% (68.1–99.8%)	44.4% (25.5–64.7%)
Group 4 > 60 years	1.4	0.652 (0.544–0.76)	34.8% (16.4–57.3%)	95.7% (78.1–99.9%)	88.9% (51.8–99.7%)	59.5% (42.1–75.2%)
Thumdiff	Median to radial comparison, thumbdiff: median latency (D1)–radial latency (D1)					
Whole cohort	1.4	0.732 (0.66–0.804)	55.1% (42.6–67.1%)	91.3% (79.2–97.6%)	90.5% (77.4–97.3%)	57.5% (45.4–69%)
Group 1 < 30 years	0.8	1 (0.0–1)	100% (15.8–100%)	100% (2.5–100%)	100% (15.8–100%)	100% (2.5–100%)
Group 2 30–39 years	0.8	0.833 (0.0–1)	66.7% (9.43–99.2%)	100% (2.5–100%)	100% (15.8–100%)	50% (1.26–98.7%)
Group 3 40–49 years	1.0	0.785 (0.555–1)	76.9% (46.2–95%)	80% (28.4–99.5%)	90.9% (58.7–99.8%)	57.1% (18.4–90.1%)
Group 4 50–59 years	1.2	0.755 (0.628–0.882)	64.3% (44.1–81.4%)	86.7% (59.5–98.3%)	90% (68.3–98.8%)	56.5% (34.5–76.8%)
Group 4 > 60 years	1.8	0.675 (0.565–0.785)	39.1% (19.7–61.5%)	95.8% (78.9–99.9%)	90% (55.5–99.7%)	62.2% (44.8–77.5%)
Ringdiff	Median to ulnar comparison at ring finger study (ringdiff): median latency (Digit IV)–ulnar latency (Digit IV)					
Whole cohort	1.7	0.631 (0.562–0.701)	30.9% (19.1–44.8%)	95.3% (84.2–99.4%)	89.5% (66.9–98.7%)	51.9% (40.4–63.3%)
Group 1 < 30 years	0.5	0.75 (0.0–1)	50% (1.26–98.7%)	100% (2.5–100%)	100% (2.5–100%)	50% (1.26–98.7%)
Group 2 30–39 years	0.5	0.667 (0.0–1)	33.3% (0.84–90.6%)	100% (2.5–100%)	100% (2.5–100%)	33.3% (0.84–90.6%)
Group 3 40–49 years	0.5	0.875 (0.715–1)	75% (34.9–96.8%)	100% (29.2–100%)	100% (54.1–100%)	60% (14.7–94.7%)
Group 4 50–59 years	1.7	0.688 (0.508–0.867)	37.5% (8.52–75.5%)	100% (29.2–100%)	100% (29.2–100%)	37.5% (8.52–75.5%)
Group 4 > 60 years	1.6	0.602 (0.484–0.721)	25% (7.27–52.4%)	95.5% (77.2–99.9%)	80% (28.4–99.5%)	63.6% (45.1–79.6%)
CSI	Combined sensory index: sum of palmdiff + thumbdiff + ringdiff					
Whole cohort	3.5	0.707 (0.632–0.783)	50.8% (38.1–63.4%)	90.7% (77.9–97.4%)	89.2% (74.6–97%)	54.9% (42.7–66.8%)
Group 1 < 30 years	1.0	1 (0.0–1)	100% (15.8–100%)	100% (2.5–100%)	100% (15.8–100%)	100% (2.5–100%)
Group 2 30–39 years	1.0	0.833 (0.0–1)	66.7% (9.43–99.2%)	100% (2.5–100%)	100% (15.8–100%)	50% (1.26–98.7%)
Group 3 40–49 years	1.2	0.833 (0.575–1)	91.7% (61.5–99.8%)	75% (19.4–99.4%)	91.7% (61.5–99.8%)	75% (19.4–99.4%)
Group 4 50–59 years	3.0	0.788 (0.669–0.908)	65.4% (44.3–82.8%)	92.3% (64–99.8%)	94.4% (72.7–99.9%)	57.1% (34–78.2%)
Group 4 > 60 years	3.5	0.71 (0.584–0.836)	54.5% (32.2–75.6%)	87.5% (67.6–97.3%)	80% (51.9–95.7%)	67.7% (48.6–83.3%)

Ms: milliseconds. ROC: receiver operator characteristic curve.

## Data Availability

All data are available upon direct request to the corresponding author.

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
