# Peer review of "Values and Diagnostic Accuracy of Electrodiagnostic Findings in Carpal Tunnel Syndrome Based on Age, Gender, and Diabetes"

_diagnostics, 2024, doi:10.3390/diagnostics14131381_

Round 1

Reviewer 1 Report

Comments and Suggestions for Authors

This study gave us the valuable information about the cut-off value in various parameter important for CTS diagnosis. Additionally, I found it helpful that the cut-off value can vary depending on age and the presence of DM. I have a few minor comments

Introduction

  • Line 60: The acronym 'CNPL' was used first without the full term. Please provide the full term as well.

Materials and Methods

  • Line 82-87: It seems that the most important criterion for diagnosing carpal tunnel syndrome is that the symptoms are primarily located in the median nerve distribution. However, this seems to be missing from the diagnostic criteria.
  • Line 164-169: For those unfamiliar with statistical techniques, it would be helpful to provide a more detailed explanation of what the coefficient value means in a linear regression equation. What does it mean for this value to be high?

Discussion

  • Line 301: “This was attributed to the fact that ultrasound cross-sectional area (CSA) is not increased in chronic or severe CTS, which is likelier to occur with DM and CTS.” → Do you have any supporting evidence for this sentence?
  • Line 319: “Other studies found no influence of age on the outcome of CTS release, and this was attributed to the exclusion of patients with DM and thyroid disease.” → This sentence needs reference.

Author Response

see attached PDF file (it contains the responses to first and second reviewer)

Reviewer 2 Report

Comments and Suggestions for Authors

the paper analyses the possible accuracy of EDx in cts depending on age, gender, diabetes. The paper is interested but limited; the inclusion criteria are broad including patients aged 14 to 80 years presenting. The exclusion criteria exclude patients with symptoms that could not be clearly classified into cts or non cts groups potentially leading to an incomplete or biased sample. The study uses a combination of clinical features (nocturnal paresthesia, aggravation by specific activities and relief by shaking the hand ) to define cts. While these criteria are supported by literature  they may not capture all cases of cts leading to potential misclassification. Furthermore the use of the eDx parameters is not clear. An extensive classification of cts severity seems missing. Finally other potential counfounders (e.g. duration of symptoms, occupational factors, other comorbidities) might also influence the results and are not adequately accounted for in the analysis 

Comments on the Quality of English Language

minor editing 

Author Response

see attached PDF file (it contains the responses to first and second reviewers)

Round 2

Reviewer 2 Report

Comments and Suggestions for Authors

All is ok

Comments on the Quality of English Language

All is ok